# Level of completion of maternity continuum of care among ever-married women: An analysis of Somalia's health and demographic survey 2020

**Adam A. Mohamed**[1,2]*, **Ayşe Akın**[1], **Sare Mihciokur**[1], **Sarp Üner**[3], **Abdi Gele**[4]

**1** Department of Public Health, Institute of Health Sciences, Başkent University, Ankara, Turkey,
**2** Department of Program Quality & Development, Save the Children International, Mogadishu, Somalia, **3** Department of Public Health, Faculty of Medicine, Lokman Hekim University, Ankara, Turkey, **4** Department of Research, Norwegian Institute of Public Health, Oslo, Norway

* 21810510@mail.baskent.edu.tr

## Abstract

### Introduction

Somalia is continuing to recover from three decades of underdevelopment, political instability, civil unrest, and protracted humanitarian crises. However, Somalia has one of the lowest maternal health indicators in the world. For instance, the maternal mortality ratio is 621 per 100,000 live births. Extra efforts are needed to improve maternal health. In this study, we aim to investigate the level of completion and coverage along the maternity continuum of care in Somalia.

### Method

The study used data from the Somalia Health and Demographic Survey 2020. We restricted our analysis to ever-married women who had a live birth in the five years preceding the survey (n = 2432). Completion of the continuum of maternity care was the outcome variable for this study. It was constructed into a binary variable with complete coded as one and incomplete coded as 0. We categorized it into three models: ANC4+ as the first model, ANC4+ & SBA as the second model, and ANC4+ & SBA & PNC as the third model.

### Results

More than half of the women (53.1%) had their most recent births at ≤19 years old. Of all the mothers (n = 2432), only 235 (9.7%) had at least four or more of the recommended antenatal care (ANC4+), and 68 (2.8%) of them utilized skilled birth attendants. Only 14 (0.6%) women received all three maternal healthcare services (ANC4+, SBA, and PNC within 48 hours). About 78.1% of the mothers did not attend any of the three CoC services.

**Data availability statement:** All relevant data are within the manuscript and its supporting information files.

**Funding:** The authors received no specific funding for this work.

**Competing interests:** The authors have declared that no competing interest exist.

## Conclusion

Maternal health care utilization decreases as they progress from ANC4+ to PNC utilization. The government and partners should design and implement strategies to improve maternal healthcare utilization specific to rural and nomads, less educated, not working, low income, and have less power in decision-making.

## 1. Background

Sub-Saharan Africa (SSA) has the highest rates of maternal mortality in the world, accounting for almost 90% of the global burden of maternal mortalities [1,2]. This high burden of maternal mortality in the region has been attributed to preventable factors such as low antenatal care (ANC) utilization, low uptake of skilled attendant delivery and postnatal care (PNC) [3]. In 2020, the MMR in the African Region was 531 deaths per 100,000 live births, accounting for 69% of global maternal deaths in 2020 [4]. From 2000 to 2020, the maternal mortality ratio dropped by about 34% worldwide. However, in many low- and middle-income countries, maternal mortality remains a significant public health problem, with nearly 94% of all maternal deaths occurring in low-resource settings to date [5]. Somalia is recovering from three decades of underdevelopment, political instability, civil unrest, and protracted humanitarian crises. The country has suffered multiple emergencies, including flooding, drought, famine, locust attacks, and other climate change shocks, which resulted in many deaths and large-scale population displacements [6].

Somalia has a population of 17 million, with 44 percent living in urban areas, 23 percent living in rural areas, 26 percent living in nomadic areas, and 9 percent living in Internally Displaced settings (IDPs). Due to limited state capacity, poverty-related deprivation, corruption, and longstanding armed conflicts, the country has one of the weakest healthcare systems in Sub-Saharan Africa. Therefore, Somalia has some of the lowest health indicators in the world [6]. The fragile health system in Somalia is shaped by various administrations that adopt different policies, priorities, and health care service approaches, often influenced by local state administrations and international paradigms and resolutions [7].

Somalia is among the 15 countries that WHO marked as very high alert countries for maternal, newborn, and under-5 deaths. Most of the causes are either preventable or treatable. Somalia has one of the worst maternal conditions in the world. For instance, the maternal mortality ratio is 621 per 100,000 live births. Poor maternal health care delivery in rural communities results in the majority of maternal, newborn, and child deaths during pregnancy, childbirth, and after delivery in Somalia [8]. This is due to the unavailability and low utilization of maternal healthcare services, including skilled birth attendance, emergency obstetric care, postnatal care, and family planning. According to a predictive analysis of the trends of maternal mortality ratio, Somalia will not meet the sustainable development goal target 3.1 of reducing the maternal mortality ratio to less than 70 per 100,000 live births. The country will not meet the targets of reducing neonatal mortality to below 12 per 1000 live births and under-5 mortality to below 25 per 1000 live births by 2030 [9]. According to Somalia's Voluntary National Reviews Report 2022 and voluntary national reviews in East African countries on Sustainable Development Goals, especially goal 3, Somalia still faces significant challenges in maternal mortality rates, which can be attributed to low uptake of antenatal care and postnatal care and a low number of deliveries at health facilities or with skilled health care providers. Somalia is unlikely to achieve the SDGs given the lack of evidence-based knowledge that informs government interventions to improve maternal and newborn health [10].

According to the SHDS, Somalia's total fertility rate is 6.9 children. Additionally, 91 percent of women interviewed in the last Somalia Health and Demographic Survey consider six or more children as their ideal family size. Age at first marriage is an important indicator of exposure to the risk of conception and childbirth, especially in a society in which all births occur within marriage. Around 16% of the ever-married reproductive-age women are married at the age of 15 years or below, while 34% are married at the age of 18 years or below. Women who marry early are known to have a higher chance of getting pregnant and having more children during their reproductive years. Women living in remote areas are less likely to receive adequate health care. This is especially true for countries with sparse skilled healthcare providers, such as Sub-Saharan Africa [11]. This is due to health system failure, which translates to inadequate quality of care, insufficient numbers of and inadequately trained health workers, shortages of essential medical supplies, and the poor accountability of health systems. Harmful gender norms and inequalities also significantly impact the prioritization of the rights of women and girls, particularly regarding their access to safe, quality, and affordable sexual and reproductive health services [12].

Continuum of Care (CoC) refers to the continuity of care throughout pregnancy, birth, and after delivery (i.e., antenatal care, skilled birth attendance, and postnatal care). Ensuring that continuity of care for maternal, newborn, and child health has become a key to improving the health of mothers, newborns, and children. The continuum of care has newly been emphasized as a core principle of programs for maternal, newborn, and child health and as a method to reduce the burden of maternal, newborn, and child deaths [13]. Primary services for a continuum of care, like antenatal care, facility delivery, and postnatal care, are recognized to reduce maternal and child mortality and morbidity in high-burden settings.

The 2023 UN Report on Trends in Maternal Mortality from 2000 to 2020 revealed that Somalia continues to experience one of the world's highest ratios of maternal mortality, with 621 maternal deaths per 100,000 live births. This is in addition to persistently poor outcomes for newborns, with 36 newborn deaths per 1000 live births and 28 stillbirths per 1000 births [14]. Maternal mortality and morbidity in Somalia are unacceptably high, with a maternal mortality rate of 621 deaths per 100,000 live births, one of the highest in the world. The 2020 Somalia Health and Demographic Survey (SHDS 2020) revealed poor ANC coverage of 31%, with only 24% of pregnant women attending at least 4 ANC visits and 21% delivering at the health facility. Moreover, care for expectant mothers throughout their pregnancy remains particularly poor, with only 32% of births attended by skilled health personnel [15].

Although the Somali government has taken a significant step to increase the availability and access to maternal care, a low proportion of mothers still utilize the services, especially in urban areas [12]. The recent Somalia Harmonized Health Facility Assessment 2022–2023 provides valuable insights into the healthcare workforce distribution across the six federal member states [16]. Physicians: three per 10,000 people (equivalent to one physician for every 3,400 people). Non-Physician Paramedical Practitioners: 1.1 per 10,000 people. Nurses: 3.9 per 10,000 people (equivalent to one nurse for every 2,500 people). Midwives: 1.5 per 10,000 people (equivalent to one midwife for every 6,500 people). When combining physicians, non-physician paramedical practitioners, nurses, nurse-midwives, and midwives, there are 7.9 health workers per 10,000 people, equivalent to one health worker for every 1,300 people [16]. Healthcare workers are often concentrated in urban areas, leaving rural communities underserved. This geographic inequality exacerbates health disparities, making it difficult for most of the mothers to access essential health services [17]. Improving the utilization of the maternity continuum of care relies on a better understanding of the barriers and gaps affecting the uptake of each service, i.e., ANC4+, SBA, and PNC. In this study, we aim to investigate the level of completion of the maternity continuum of care in Somalia.

## 2. Methods

### 2.1. Hypothesis structure used for the completion of the continuum of care

This study used population level continuum of care framework based on integrated service delivery by using three major packages for maternal healthcare utilization such as: Antenatal Care (ANC+4), Skilled Birth Attendant (SBA), and mother's Post-natal Care (PNC). Antenatal Care (ANC) is defined as care that pregnant women received from skilled healthcare providers during pregnancy more than 4 times. The information regarding ANC+4 was obtained from "number of ANC visits during pregnancy or how many times you received antenatal care during this pregnancy?". Facility-based delivery is defined as the delivery of pregnant women at a health facility attended by healthcare professionals (doctor, midwives, and nurses). The information regarding skilled birth attendant or institutional delivery (ID) was obtained from the question: "Place of delivery or Where did you give birth to the last child?" and women were considered to have institutional delivery if they had a delivery at hospital, clinic, or any other health facility or if their delivery was assisted by a skilled birth attendant. Furthermore, information on PNC of the mother was obtained from the question: "Respondent's health checked before discharge, or Did anyone check on your health while you were still in the facility?". The study followed the below Fig 1 hypothesized structural relation for continuum of care.

### 2.2. Sampling procedure

The sample for the SHDS was designed to provide estimates of key indicators for the country as a whole, for each of the eighteen pre-war geographical regions, which are the country's first-level administrative divisions, as well as separately for urban, rural, and nomadic areas. The SHDS followed a stratified multi-stage probability cluster sample design. The sample design in urban and rural areas was a three-stage stratified cluster sample design, while in nomadic areas, the design was a two-stage stratified cluster sample design. With the exception of the Banadir region, which is considered fully urban, each region was stratified into urban, rural, and nomadic areas, yielding a total of 55 sampling strata. All three strata of Lower Shabelle and Middle Juba regions, as well as the rural and nomadic strata of the Bay region, were completely excluded from the survey due to security reasons. A final total of 47 sampling strata formed the sampling frame. Through the use of up-to-date, high-resolution satellite imagery, as well as on-the-ground knowledge of staff from the respective ministries of planning, all dwelling structures were digitized in urban and rural areas.

Enumeration Areas [18] were formed onscreen through a spatial count of dwelling structures in Geographic Information System (GIS) software. Thereafter, a sample ground verification of the digitized structures was carried out for large urban and rural areas, and

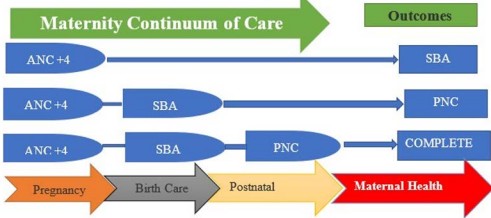

**Fig 1. Hypothesis structural relationships for CoC, Adapted from Kerber (Lancet) and Ajinkya (India) as in the references.**

necessary adjustments were made to the frame. Each EA created had a minimum of 50 and a maximum of 149 dwelling structures. A total of 10,525 EAs were digitized: 7,488 in urban areas and 3,037 in rural areas. However, because of security and accessibility constraints, not all digitized areas were included in the final sampling frame, and 9,136 EAs (7,308 in urban and 1,828 in rural) formed the final frame. The nomadic frame comprised an updated list of temporary nomadic settlements (TNS) obtained from the nomadic link workers tied to these settlements. A total of 2,521 TNS formed the SHDS nomadic sampling frame.

## 2.3. Data source and study participants

This study used data from the 2020 Somalia Health and Demographic Survey, a nationally representative household survey designed to collect, analyze, and disseminate demographic data on reproductive health, maternal and child mortality, family planning and fertility, nutrition, and health care utilization in Somalia. The main objective of the Somali Health and Demographic Survey was to provide evidence on the health and demographic characteristics of the Somali population that will guide the development of programs and formulation of effective policies. This nationally representative survey used a multistage cluster sampling design to collect data on reproductive health, maternal and child mortality, family planning and fertility, nutrition, and health utilization in Somalia.

About 11,884 ever-married women of reproductive age (15–49 completed the interviews. Our study focused on ever-married women who had a live birth in the five years preceding the survey (n = 2432), as shown in Fig 2 below. The survey further restricted the women's most recent live births in the recall period. Nine thousand four hundred seventy were not eligible due to either not having children under 5 years, incomplete data in the maternal and child health questions, or not fitting the targets, and the analysis removed them. Marriage is associated with childbearing; a woman can have a child only if married, as accepted in the community. In the Somalia context, all women who have children had marriage previously, which means women cannot have a child without legal marriage. That is why we restricted our study to only the ever-married women.

9,470 women were not eligible due to either needing to have children less than 5 years of age, incomplete data in the maternal and child health questions, or not fitting the targets, and we removed them during the data analysis.

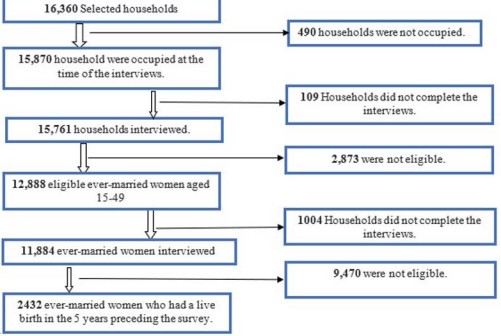

**Fig 2. Selection process of eligible women aged 15–49 years old using Somalia Health and Demographic Survey 2020.**

## 2.4. Data analysis

We used STATA software (version 18) and sampling weight for all analyses with the survey data analysis command (say) to account for the cluster survey design and missing responses. The study did not proportionally sample urban, rural, and nomadic domains; hence, urban overrepresentation may occur. Many of the population refused to finalize the questionnaires, especially the maternal and child health section, causing incomplete interviews. The SHDS data are missing because some aspects of the population were not covered. The SHDS data in Somalia was not meant to track detailed elements of maternity care, which makes it challenging to analyze the maternity continuum of care. We checked the data for completeness and then cleared it. We then conducted a descriptive analysis (number, frequency, and percentage) to summarize the characteristics of the study population and each variable. We assessed the level of coverage for each maternal health service (ANC, SBA, and PNC) separately and then together as a combined continuum of maternity care. We used the 48-hour postpartum period since it is the critical period window that can emphasize the most urgent phase for maternal and newborn interventions. Our data are almost the same for 48 hours and 42 days (low utilization of PNC services beyond 48 hours). We also compared our study with other studies that used 48-hour PNC visits. Variance Inflation Factor was used to test the presence of collinearity between the independent variables using STATA 18.

## 3. Results

### 3.1. Characteristics of the study participants

Table 1 summarizes the background characteristics of ever-married mothers included in this study. The mean age of the women in the study was 22.8 years (SD = 4.2), with more than half of the women (53.1%) had their most recent births at the age of ≤19 years, while 90.2% of the women in the study were married during the survey period. Above sixty percent of the women lived in urban areas, and seventy-seven had no education. 96.1% of the women were not working, 74.5% had 1–2 children, and 90.6% had not accessed mass media (radio exposure). Wealth quintile distribution among the households was almost homogenous, with the highest being the poorest (25.5%) and the lowest being the middle group (16.7%), as shown in Table 1 below.

### 3.2. The overall use of maternal health services in Somalia

Table 2 shows the descriptive analysis indicating that Somalia did not achieve good antenatal care, institutional delivery, and postnatal care coverage. Around **two-thirds** of the study participants (66.2%) did not receive antenatal care visits, and only 24.2% received ANC between 1–3 trips, whereas 9.7% of the women in the study attended the recommended four or more antenatal care (ANC4+) visits. Skilled providers attended less than 14% of the deliveries. For postnatal care, only 2.7% of the women had a PNC check within 48 hours after delivery, regardless of their place.

**Maternity Continuum of Care:** Of all the mothers (n = 2432), only 235 (9.7%) had at least four or more of the recommended antenatal care (ANC4+) visits during their recent pregnancy and among this, only nine mothers live in nomadic settings. Out of the women who have received four or more ANC care, only 68 (2.8%) of them utilized skilled birth attendants in their most recent pregnancies. Regarding the continuum of maternal health services, only 14 (0.6%) of women had received all three maternal healthcare services (ANC4+, SBA, and PNC within 48 hours) as shown in the below Figs 3 and 4, respectively.

**Table 1. Socio-demographic characteristics of ever-married women who had at least one live birth in the five years preceding the survey (weighted sample size = 2432 & unweighted sample size = 2414).**

| Variable | Categories | Weighted Number (2432) | Percentage (%) |
|---|---|---|---|
| **Maternal age at birth (years)** | <20 | 1,291 | 53.1 |
|  | 20–34 | 1,125 | 46.3 |
|  | 35–49 | 16 | 0.6 |
| **Birth Order** | 1 | 906 | 37.3 |
|  | 2 | 908 | 37.3 |
|  | 3 | 525 | 21.6 |
|  | Four or more | 93 | 3.8 |
| **Residence** | Urban | 1,475 | 60.6 |
|  | Rural | 653 | 26.9 |
|  | Nomadic | 304 | 12.5 |
| **Marital Status** | Married | 2,193 | 90.2 |
|  | Divorced/widowed | 239 | 9.8 |
| **Mother's Education** | No Education | 1,872 | 77.0 |
|  | Primary | 370 | 15.2 |
|  | Secondary & above | 190 | 7.8 |
| **Mother's Employment** | Working | 96 | 3.9 |
|  | Not working | 2,336 | 96.1 |
| **Number of Children** | 1–2 | 1,813 | 74.5 |
|  | 3–4 | 598 | 24.6 |
|  | >=5 | 21 | 0.9 |
| **Decision-making power in determining one's healthcare** | Women or jointly | 1,285 | 52.8 |
|  | Husband or others | 1,147 | 47.2 |
| **Exposure to Radio** | Yes | 229 | 9.4 |
|  | No | 2,203 | 90.6 |
| **Wealth Quintile** | Poorest | 621 | 25.5 |
|  | Poorer | 486 | 20.0 |
|  | Middle | 406 | 16.7 |
|  | Richer | 454 | 18.7 |
|  | Richest | 465 | 19.1 |

**Table 2. Number and Percentage of mothers and their use of maternal health services.**

| Characteristics | Categories | Weighted Number (2432) | Percentage (%) |
|---|---|---|---|
| **ANC visits** | No ANC | 1,608 | 66.2 |
|  | 1-3ANC | 589 | 24.2 |
|  | ANC4+ | 235 | 9.7 |
| **Delivered by SBA** | Skilled provider | 327 | 13.4 |
|  | Unskilled provider | 1,478 | 60.8 |
|  | No one | 47 | 1.9 |
|  | Don't know/missing | 580 | 23.8 |
| **PNC visits** | First 48hrs | 67 | 2.7 |
|  | More than 48hrs | 2 | 0.1 |
|  | No PNC/Don't know | 2,363 | 97.2 |

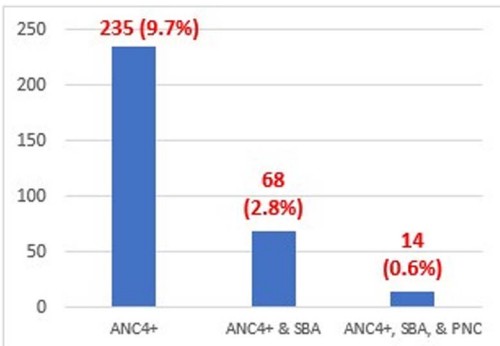

**Fig 3. Maternity Continuum of Care.**

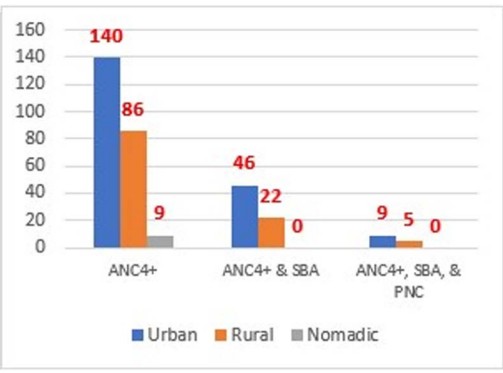

**Fig 4. Maternity Continuum of Care by residence.**

**Table 3. Percentage distribution of the three maternal health services received by women. + Received the service; − Did not receive the service.**

| Pathway | ANC4+ | SBA | PNC within 48hrs | Frequency (Weighted Number) | Percentage |
|---|---|---|---|---|---|
| 1 | − | − | − | 1,899 | 78.1 |
| 2 | + | − | − | 164 | 6.7 |
| 3 | + | + | − | 54 | 2.2 |
| 4 | − | − | + | 39 | 1.6 |
| 5 | − | + | − | 248 | 10.2 |
| 6 | − | + | + | 11 | 0.5 |
| 7 | + | − | + | 3 | 0.1 |
| 8 | + | + | + | 14 | 0.6 |
| **Total** | | | | **2,432** | **100%** |

## 3.3. Pathways of maternal healthcare use

Considering the study's three main outcomes (ANC4+, SBA, and PNC), we created eight different pathways or combinations of maternal health service utilization, i.e., from not receiving any of the three maternal health services to completing the use of all three maternal health services (Table 3). More than two-thirds of the mothers in the study (78.1%) did not receive

any of the three maternal health services as shown in pathway 1. Notably, only 14 mothers (0.6%) received all of the three maternal health services along the continuum (pathway 8). Three groups that attended any two of the three maternal health services (pathways 3, 6, and 7) accounted for 2.2%, 0.5%, and 0.1%, respectively. Lastly, the three groups that utilized only one of the three maternal services (pathways 2, 4, and 5) accounted for 6.7%, 1.6%, and 10.2%, respectively.

Table 3 above shows the combination of maternal health services (three major services) that mothers either received or did not receive. It is summarizing the three major outcome variables of the study, i.e., antenatal care of four times or above, skilled birth attendant, post-natal care within 48 hours after delivery.

## 4. Discussion

This study assessed the maternity continuum of care for the first time in Somalia. Access and use of maternity care services during pregnancy, childbirth, and the postnatal period from skilled providers are essential for the survival and wellbeing of the mother and newborn. It is particularly critical in settings where teenage pregnancy is common, such as Somalia. Our study states that 53% of women gave birth to their first child at the age of ≤19, while the completion of ANC4 was only 9.7%. This rate is much higher than the completion of ANC4, which is 59.9% in Uganda [19] And that of Ethiopia, which is 33% [20]. Protracted conflicts and instability may explain why women in Somalia have a low prevalence of ANC4. Armed conflict has been described as an important contributor to persistent excess maternal and child deaths while it can severely reduce access to maternal health services and thus lead to poor maternal health outcomes [21]. The effect of the three decades of conflict on the access to health care in Somalia is exacerbated by the pervasive corruption in the health sector, given the fact that systemic corruption and the large-scale misappropriation of state funds is the norm in Somalia [22]. In Somalia, like many other countries, corruption in the health sector ranges from smaller-scale acts by doctors and nurses who charge "informal payments" to larger-scale acts at the ministerial level, when people in power employ incompetent relatives or redirect resources away from those who need it for their own benefit [23]. Hence, countries with high levels of corruption spend less on health care [24]. In addition, high levels of corruption correlate with higher infant and child mortality rates [25]. Improving maternal healthcare in Somalia requires political and economic commitment, and genuine leadership is critical to achieving this agenda.

Our study finds that the overall completion of maternity continuum of care is less than 1%. This is much lower than that reported in Gondor Ethiopia 47% [26]. This could be due to the efforts made by the Ethiopian government to in the recent years to strengthen maternity health care access.

Our study shows that only 13.4% of women in Somalia gave birth with the help of skilled health providers. A prior study on 29 Sub-Saharan African countries found an average proportion of women who had skilled assistance during delivery was 75.3%, ranging from 38.4% in Chad to 93.7% in Rwanda [3]. Skilled attendants have a positive contribution to the reduction of maternal and newborn mortality and morbidity. A 40% decline in maternal deaths in SSA occurred between 2000 and 2017 was attributed, by WHO, to utilizing skilled birth attendants at delivery [27]. The fact that only 13.4% of women in Somalia are using the services of a skilled birth attendants, indicated attendants indicates that although utilization of the services of SBAs in SSA appears high, it is extremely low in Somalia. The SBA prevalence in Somalia can be improved by adopting prevailing successful interventions of countries with SBA successes, such as Rwanda, whilst considering contextual variations [28]. Our study finds that the overall completion of the maternity continuum of care is less than 1%. This is much

lower than that reported in Gondor, Ethiopia 47% [26]. This could be due to the efforts made by the Ethiopian government to in the recent years to strengthen maternity health care access.

Our study found that only 2.8% of Somali women have received postnatal care. Receiving postnatal care after delivery is vital for women and the baby due to the fact that over 65% of maternal and neonatal deaths occur during the first 42 days of postpartum and during the first 7 days of life, respectively. A study on postnatal care utilization in 36 sub-Saharan African countries found a prevalence of PNC of 52.48% [95% CI: 52.33, 52.63], ranging from 73.51% in Central Africa region to 31.71% in the Eastern Africa Region [29]. A prior study found an association of armed conflicts with an increase of 36.9 maternal deaths per 100,000 live births and an increase of 2.8 infant (under 1 year old) deaths per 1,000 live births [21]. The continuity of maternity health care services is the care that a woman uses the three recommended cares of antenatal care (ANC), skill birth attendant (SBA), and postnatal care (PNC). Our study shows that only 0.6% of the women in Somalia have completed the three services: ANC4, SBA, and PNC, while only 2.8% have completed ANC4 and SBA. The prevalence of maternal continuum of care in Somalia, which is stated by our study (0.6%), is far lower than the frequency of the continuum of care in Ethiopia (47%) [26] and is even over 15 times lower than the continuum of care of rural women in Ethiopia (13%) [30]. Further, our study shows a significant inequity in the continuum of care by a residence with zero continuum of care among women in pastoral communities that constitute 26% of the total population in Somalia. A recent study in Somalia stated that pastoralist women have limited access to maternal health care, which supports our findings [12]. The inequity between urban and rural women in utilizing maternal health care services is prevalent across Africa. A prior study of 27 African countries showed a prevalence of ANC utilization of 34.7% among urban women and 22.4% among in rural areas, while SBA prevalence was nearly 90% and 69% in urban women and rural women, respectively [31].

This study has both strength and limitations. The main strength of this study is the use of weighted nationally representative data with a large sample, which makes it representative at national level. The SHDS data used a multi-stage cluster sampling design, and some groups or demographic categories might be over sampled hence, we weighed the SHDS data during analysis.

The current SDHS did not collect information on type of provider for each visit, therefore, we couldn't determine the type of provider for each visit. Further, approximately 2.6 million Somalis are currently displaced within their own country and the SHDS 2020 did not include the IDP domain in the data.

## Conclusion and recommendations

The prevalence of completion of the maternity continuum of care was found to be lower than in any country in Africa. Maternal health care utilization decreases as they progress from ANC4+ to PNC utilization. Inequity in the continuum of care is a chronic problem in Somalia, where pastoralist women, who constitute 26% of the population, have zero completion of the continuum of care. Health inequalities are avoidable because they are rooted in political and social decisions. Political and economic commitment is required to improve access to maternal health care in the country. The findings of our study indicate that if the government does not target the rural and pastoralist communities, women's retention of health services if existing, would be minimal.

To reduce health inequalities between urban and pastoralists in Somalia, we need to act across a range of health-policy areas, including policies to improve access to maternal health care in pastoralist settings in the country. The government and partners should design and implement strategies to improve maternal healthcare utilization specific to rural and nomads,

less educated, not working, low income, and have less power in decision-making. The government of Somalia should invest in and increase the number of certified skilled birth attendants, i.e., nurses, midwives, and obstetricians, to provide safe and competent care during childbirth. To attract and retain skilled birth attendants in remote and rural areas, the government and its partners should implement incentive programs designed for professional and certified midwives and obstetricians and should train and deploy community health extension workers in rural areas to provide basic maternal health education, identify pregnant women, and support and encourage them to seek health care.

## Supporting information

**S1 Data.  Somali Health and Demographic Survey data set (SHDS).**
(SAV)

## Author contributions

**Conceptualization:** Adam A. Mohamed, Ayşe Akın, Abdi Gele.

**Data curation:** Adam A. Mohamed.

**Formal analysis:** Adam A. Mohamed.

**Investigation:** Adam A. Mohamed.

**Methodology:** Adam A. Mohamed, Abdi Gele.

**Supervision:** Ayşe Akın.

**Writing – original draft:** Adam A. Mohamed, Abdi Gele.

**Writing – review & editing:** Ayşe Akın, Sare Mihciokur, Sarp Üner, Abdi Gele.

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
