## [Decision Letter · Decision Letter 0]

9 Sep 2024

PGPH-D-24-00010

Level of Completion of Maternity Continuum of Care among ever-married women: An analysis of Somalia Health and Demographic Survey 2020

Dear Dr. Mohamed,

Thank you for submitting your manuscript to PLOS Global Public Health. After careful consideration, we feel that it has merit but does not fully meet PLOS Global Public Health’s publication criteria as it currently stands. Therefore, we invite you to submit a revised version of the manuscript that addresses the points raised during the review process.

The manuscript has been evaluated by two reviewers, and their comments are available below.

The reviewers have raised a number of minor concerns. Could you please carefully revise the manuscript to address all comments raised?

We look forward to receiving your revised manuscript.

Kind regards,

Johanna Pruller, Ph.D.

Staff Editor

Journal Requirements:

-https://doi.org/10.3389/fpubh.2022.1026236

In your revision ensure you cite all your sources (including your own works), and quote or rephrase any duplicated text outside the methods section. Further consideration is dependent on these concerns being addressed.

3. We have amended your Competing Interest statement to comply with journal style. We kindly ask that you double check the statement and let us know if anything is incorrect. 

4. Please provide a/amend your detailed Financial Disclosure statement. This is published with the article. It must therefore be completed in full sentences and contain the exact wording you wish to be published.

**Please only choose the relevant sentences from below**

1. Please clarify all sources of funding (financial or material support) for your study. List the grants (with grant number) or organizations (with url) that supported your study, including funding received from your institution. 

2. State the initials, alongside each funding source, of each author to receive each grant.

3. State what role the funders took in the study. If the funders had no role in your study, please state: “The funders had no role in study design, data collection and analysis, decision to publish, or preparation of the manuscript.”

4. If any authors received a salary from any of your funders, please state which authors and which funders.

5. Please upload your manuscript file "CoC Manuscript_first_draft (003).docx 16.01.2024.pdf" as a .doc, .docx or .rtf file.

6. Please provide separate figure files in .tif or .eps format.

7. In the online submission form, you indicated that "All the data related to this manuscript are available upon request". 

3. Uploaded as supplementary information.

Additional Editor Comments (if provided):

Reviewers' comments:

Reviewer's Responses to Questions

**Comments to the Author**

1. Does this manuscript meet PLOS Global Public Health’s publication criteria? Is the manuscript technically sound, and do the data support the conclusions? The manuscript must describe methodologically and ethically rigorous research with conclusions that are appropriately drawn based on the data presented.

Reviewer #1: Yes

Reviewer #2: Yes

2. Has the statistical analysis been performed appropriately and rigorously?

Reviewer #1: I don't know

Reviewer #2: Yes

3. Have the authors made all data underlying the findings in their manuscript fully available (please refer to the Data Availability Statement at the start of the manuscript PDF file)?

Reviewer #1: Yes

Reviewer #2: Yes

4. Is the manuscript presented in an intelligible fashion and written in standard English?

Reviewer #1: Yes

Reviewer #2: Yes

5. Review Comments to the Author

Reviewer #1: The manuscript, as a descriptive study, addresses a critical public health issue in Somalia, and meets PLOS Global Health criteria for publication. However, given the richness of the available data, incorporating a more in-depth inferential analysis could provide valuable insights by identifying factors or predictors associated with the outcome (maternal continuum of care).

Reviewer #2: comments to the author

The authors pic an important topic in the field of public health entitled “Level of Completion of Maternity Continuum of Care among ever-married women: An analysis of Somalia Health and Demographic Survey 2020” particularly as the authors described in the country which has the highest maternal mortality in Sub-Saharan Africa, Somalia. thank you for submitting such important and timely title for the journal. However, I have some comments and concerns to enhance the quality of this manuscript before publication.

# General typographical comments

# Abstract section: The sentence “This study used the first ever DHS data in Somalia and id not get any funding” needs attention and rewrite with correct spelling

# it would have been better if the authors have been inserted page numbers and line numbers

#Affiliations

Author 2 and author 3 should write their department first in the affiliation

# Methods: in your selection procedure it shows that “15,761 households interviewed” and “2,873 were not eligible.” After that it stated and showed that “12,888 eligible ever-married women aged 15-49” however, after interview “9,470 were not eligible” why? the authors shall clearly show the selection process for the scientific community it is confusing

# In your Dependent and independent variables:

Why did the authors list or display the independent and dependent variables? Since you have already discussed and reported the descriptive statistics, why don't you analyze the associations—at the very least, the most basic one, the chi square association—in this case?

Describe the dependent variable and the metrics used to measure it. For instance, what were the metrics used to evaluate the prenatal, institutional delivery, and postnatal care for each? The authors limit postnatal care, in particular, to 48 hours after delivery; however, they do not specify the number of visits that should take place during this time. The authors should also provide an explanation for their decision to use 48 hours rather than the 42-day postpartum period and at least 4 PNC visits.

# Discussion section

“A prior study on 29 Sub-Saharan African countries found an average proportion of women who had skilled assistance during delivery was 75.3%, ranging from 38.4% in Chad to 93.7% in Rwanda However, our study shows that only 13.4% of women in Somalia gave birth with the 10 help of skilled health provider.” The author attempts to compare their findings with those of previous studies. It would be preferable to compare with other comparable descriptive research because these studies are distinct from one another and cannot make sense when descriptive frequency-based results are compared to proportion-based findings.

“The low utilization of SBA in Somalia can be improved by adopting prevailing successful interventions of countries with SBA successes such as Rwanda, whilst taking into account of contextual variations” needs reference citation

The authors stated that” The prevalence of maternal continuum of care in Somalia which is stated by our study (0.6%)” can the author put the confidence interval for this prevalence? I think it is a descriptive result and frequency not a “prevalence” furthermore, the authors stated that “The prevalence of maternal continuum of care in Somalia which is stated by our study (0.6%), is far much lower than the prevalence of continuum of care in Sub-Saharan Africa (25.0%), which varied from 17.9.0% in East Africa to 7,418 (51.5% in Southern Africa (27). It is also much lower than the 35.81% pooled prevalence of completion of the maternity continuum of care among 33 sub-Saharan African countries” It is not clear how to compare this one descriptive result with the pooled prevalence; instead, it would be appropriate to compare and discuss the studies with other comparable papers and setting outcomes.

I appreciate the authors' initiative and motivation, especially with regard to the crucial public health issue of the maternal health continuum of care in one of the areas with the highest rates of maternal death. However, the writers should reevaluate their work in order to improve the quality of the report.

6. PLOS authors have the option to publish the peer review history of their article (what does this mean?). If published, this will include your full peer review and any attached files.

**Do you want your identity to be public for this peer review?** For information about this choice, including consent withdrawal, please see our Privacy Policy.

Reviewer #1: No

Reviewer #2: No

---

## [Decision Letter · Decision Letter 1]

29 Oct 2024

PGPH-D-24-00010R1

Level of Completion of Maternity Continuum of Care among ever-married women: An analysis of Somalia Health and Demographic Survey 2020

Dear Dr. Adam Abdulkadir Mohamed,

Thank you for submitting your manuscript to PLOS Global Public Health. After careful consideration, we feel that it has merit but does not fully meet PLOS Global Public Health’s publication criteria as it currently stands. Therefore, we invite you to submit a revised version of the manuscript that addresses the points raised during the review process.

We look forward to receiving your revised manuscript.

Kind regards,

Damen Haile Mariam, MD, MPH, PhD

Academic Editor

Journal Requirements:

Additional Editor Comments (if provided):

Reviewer 1:

- The manuscript, as a descriptive study, addresses a critical public health issue in Somalia, and meets PLOS Global Health criteria for publication. However, given the richness of the available data, incorporating a more in-depth inferential analysis could provide valuable insights by identifying factors or predictors associated with the outcome (maternal continuum of care).

Reviewer 2:

- General typographical comments:

- Abstract section: The sentence “This study used the first ever DHS data in Somalia and id not get any funding” needs attention and rewrite with correct spelling.

- It would have been better if the authors have been inserted page numbers and line numbers

- Affiliations:

- Author 2 and author 3 should write their department first in the affiliation.

Methods:

- In your selection procedure it shows that “15,761 households interviewed” and “2,873 were not eligible.” After that it stated and showed that “12,888 eligible ever-married women aged 15-49”

however, after interview “9,470 were not eligible” why? the authors shall clearly show the selection process for the scientific community it is confusing

- In your Dependent and independent variables: Why did the authors list or display the independent and dependent variables? Since you have already discussed and reported the descriptive statistics,

why don't you analyze the associations—at the very least, the most basic one, the chi square association—in this case?

Describe the dependent variable and the metrics used to measure it. For instance, what were the metrics used to evaluate the prenatal, institutional delivery, and postnatal care for each? The

authors limit postnatal care, in particular, to 48 hours after delivery; however, they do not specify the number of visits that should take place during this time. The authors should also provide an

explanation for their decision to use 48 hours rather than the 42-day postpartum period and at least 4 PNC visits.

- Discussion section

- “A prior study on 29 Sub-Saharan African countries found an average proportion of women who had skilled assistance during delivery was 75.3%, ranging from 38.4% in Chad to 93.7% in Rwanda.

However, our study shows that only 13.4% of women in Somalia gave birth with the 10 help of skilled health provider.” The author attempts to compare their findings with those of previous

studies.

It would be preferable to compare with other comparable descriptive research because these studies are distinct from one another and cannot make sense when descriptive frequency-based results

are compared to proportion-based findings.

- “The low utilization of SBA in Somalia can be improved by adopting prevailing successful interventions of countries with SBA successes such as Rwanda, whilst taking into account of contextual

variations” needs reference citation

- The authors stated that” The prevalence of maternal continuum of care in Somalia which is stated by our study (0.6%)” can the author put the confidence interval for this prevalence? I think it is a

descriptive result and frequency not a “prevalence” furthermore, the authors stated that “The prevalence of maternal continuum of care in Somalia which is stated by our study (0.6%), is far much

lower than the prevalence of continuum of care in Sub-Saharan Africa (25.0%), which varied from 17.9.0% in East Africa to 7,418 (51.5% in Southern Africa (27). It is also much lower than the

35.81% pooled prevalence of completion of the maternity continuum of care among 33 sub-Saharan African countries” It is not clear how to compare this one descriptive result with the pooled

prevalence: instead, it would be appropriate to compare and discuss the studies with other comparable papers and setting outcomes.

Reviewer 3:

- General comment:

- The description presented in this study is similar with the reports of SDHS. It does not add new knowledge to the available evidence. However, the authors' interest with regard to the crucial public

health issue of continuum of care for improving maternal health is appreciated.

- Abstract:

- Introduction: Somalia has one of the worst maternal conditions in the world with maternal mortality ratio of 692 per 100,000 live births. If maternal mortality is described, what is the importance

of including a phrase “maternal conditions”? - Rephrase the sentence.

- Methods: There are inconsistences in the outcome stated in the objective and method. The outcome variable stated in the method section is only completion of maternal continuum of care.

However, the aim of the study is to investigate the coverage and the level of completion of continuum of maternity care. This needs aligning.

- Results and Conclusion: The findings presented in results section had repetition and recommendation given in the conclusion section seems generic than based on the findings. Consider revision.

- Introduction:

- The introduction section was more focused on status of maternal health in Somalia than utilization of continuum of care.

- Concept of Continuum of care was not well presented. The authors need to provide definition they used for “Continuum of care”, from available definitions in the literature.

- A paragraph written on continuum of care has redundancy of phrases.

- Limited evidence was presented on status utilization of continuum of care (globally, regionally or nationally). The gap that was addressed by this study was not clearly presented. This section

needs a major revision including addition of literature review and revision of the available evidence.

- Methods:

- It is very brief and not well described.

- Sampling procedure of the study participants was not well described.

- The procedure followed to access the data was not described and may have ethical concerns.

- Why were 9470 women not eligible in Figure 1? - Include the reason and make it self-explanatory.

- Methods used for management of missed data was not explained.

- What is the implication of this sentence «Marriage is associated with childbearing and women can have a child only if she is married as accepted in the community”? How was marriage defined in

the context of this sentence? What is the relation of marriage with child-birth? The authors have to describe it in detail.

- Results:

- Use of terms; prevalence, proportion, inequality including bracket were not appropriately used.

- Use of ANC versus antenatal care, SBA versus skilled birth attendant, and PNC versus postnatal care needs to be standardized/uniform.

- Table headings are placed wrongly.

- Discussion:

- Did not follow the standard ways of presenting major findings. The findings were not discussed with relevant evidence from the literature

- Many limitations in the methods section were not mentioned under the limitations section.

- Some information is redundant and can be taken out or shortened.

- Conclusion:

- Not comprehensively presented and it needs to be summarized

- Recommendations are generic which are not based on the finding of the study.

- References:

- Check referencing style - for instance, Ref.no. 5.

Reviewers' comments:

Reviewer's Responses to Questions

**Comments to the Author**

1. If the authors have adequately addressed your comments raised in a previous round of review and you feel that this manuscript is now acceptable for publication, you may indicate that here to bypass the “Comments to the Author” section, enter your conflict of interest statement in the “Confidential to Editor” section, and submit your "Accept" recommendation.

Reviewer #3: (No Response)

2. Does this manuscript meet PLOS Global Public Health’s publication criteria? Is the manuscript technically sound, and do the data support the conclusions? The manuscript must describe methodologically and ethically rigorous research with conclusions that are appropriately drawn based on the data presented.

Reviewer #3: No

3. Has the statistical analysis been performed appropriately and rigorously?

Reviewer #3: No

4. Have the authors made all data underlying the findings in their manuscript fully available (please refer to the Data Availability Statement at the start of the manuscript PDF file)?

Reviewer #3: No

5. Is the manuscript presented in an intelligible fashion and written in standard English?

Reviewer #3: No

6. Review Comments to the Author

Reviewer #3: Review comments

General comment: The language is not clear and it needs revision

The description presented in this study is similar with the reports of SDHS. It does not add new knowledge to the available evidences. However, the authors' interest with regard to the crucial public health issue of continuum of care for improving maternal health is apprecieated.

Title: Level of Completion of Maternity Continuum of Care among ever-married women: An analysis of Somalia Health and Demographic Survey 2020

Maternity Continuum of Care use among ever-married women: Analysis of Somalia Health and Demographic Survey 2020

Abstract:

Introduction: Somalia has one of the worst maternal conditions in the world with maternal mortality ratio of 692 per 100,000 live births. If maternal mortality is described what is the importance to include a phrase “maternal conditions”? Rephrase the sentence.

Method:

There is inconsistences of outcome stated in the objective and method. The outcome variable stated in the method section is only completion of maternal continuum of care. However, the aim of the study is to investigate the coverage and the level of completion of continuum of maternity care. This needs aligning.

Results and Conclusion:

• The findings presented in results section had repetition and recommendation given in the conclusion section seems generic than based on the findings. Consider revision.

Introduction

• The introduction section was more focused on status of maternal health in Somalia than utilization of continuum of care.

• Concept of Continuum of care was not well presented. The authors need to provide definition they used for “Continuum of care”, from available definitions in the literature.

• A paragraph written on continuum of care has redundancy of phrases.

• Limited evidences were presented on status utilization of continuum of care (globally, regionally or nationally).

• The gap that was addressed by this study was not clearly presented. This section needs a major revision including addition of literature review and revision of the available evidences.

Method

• It is very brief and not well described.

• Sampling procedure of the study participants was not well described.

• The procedure followed to access the data was not described and may have ethical concerns

• Inclusion and exclusion criteria was well describe

• Why 9470 women were not eligible in Figure 1? Include the reason and make it self explanatory

• Methods used for management of missed data was not explained.

• What is the implication of this sentence «Marriage is associated with

childbearing and women can have a child only if she is married as accepted in the community”? How was marriage defined in the context of this sentence? What is the relation of marriage with child birth? The authors have to describe it in detail.

• Analysis section

Results

• Use of terms; prevalence, proportion, inequality including bracket were not appropriately used. Suggestion use a space to separate the two

• Tenses - mixed used of present and past tenses, mainly in findings, and discussion which needs more consistency and appropriate use

• Use of ANC v antenatal care, SBA V skilled birth attendant, and PNC V postnatal care needs to be standardized/uniform – a mix of full wording and abbreviation; a mix of upper and lower case used

• Please avoid “i.e,n’t” such abbreviations – please use full wording, ‘that is’ , not, etc

• Tables headings are placed wrongly

• Good to present data in table 3 in text for better understanding

Discussion

• Did not follow the standard ways of presenting major findings.

• The findings were not discussed with relevant evidences from the literature

• Missed information and incompleteness of the data was not presented as limitation

• Reference missing for some statements based on literature.

• The study has many limitations in methods but this was not described well under limitations section

• Some of the sentence require reference

• Some information is redundant and can be cut or shortened to be a part of the next sentence

Conclusion

• Not comprehensively presented and it needs to be summarized

• Recommendations are generic which are not based on the finding of the study. Please consider including recommendation based on the findings of actual data in the text.

References

Check referencing style like Ref.no. 5

7. PLOS authors have the option to publish the peer review history of their article (what does this mean?). If published, this will include your full peer review and any attached files.

**Do you want your identity to be public for this peer review?** For information about this choice, including consent withdrawal, please see our Privacy Policy.

Reviewer #3: No

---

## [Editor Report · Decision Letter 2]

3 Dec 2024

Level of Completion of Maternity Continuum of Care among ever-married women: An analysis of Somalia Health and Demographic Survey 2020

PGPH-D-24-00010R2

Dear Dr Mohamed,

We are pleased to inform you that your manuscript 'Level of Completion of Maternity Continuum of Care among ever-married women: An analysis of Somalia Health and Demographic Survey 2020' has been provisionally accepted for publication in PLOS Global Public Health.

Best regards,

Damen Haile Mariam, MD, MPH, PhD

Academic Editor